# Acute Ischaemic Mitral Valve Regurgitation

**DOI:** 10.3390/jcm11195526

**Published:** 2022-09-21

**Authors:** Breda Hennessey, Nestor Sabatovicz, Maria Del Trigo

**Affiliations:** 1Department of Cardiology, Hospital Clínico San Carlos, 28040 Madrid, Spain; 2Department of Cardiology, Cardiocentro Cirurgia Cardiovascular, Brasilia 70000-000, Brazil; 3Department of Cardiology, Hospital Universitario Puerta de Hierro de Majadahonda, C/Manuel de Falla 1, 28222 Majadahonda, Spain

**Keywords:** acute mitral regurgitation, acute myocardial infarction, percutaneous edge-to edge repair

## Abstract

Acute ischaemic mitral regurgitation (IMR) is an increasingly rare and challenging complication following acute myocardial infarction. Despite recent technical advances in both surgical and percutaneous interventions, a poor prognosis is often associated with this challenging patient cohort. In this review, we revisit the diagnosis and typical echocardiographic features, and evaluate current surgical and percutaneous treatment options for patients with acute IMR.

## 1. Introduction

Mitral regurgitation (MR) is the second most common valvular disease in Europe [1]. Primary MR is caused by intrinsic valvular disease, with secondary MR developing due to changes in the geometry of the left ventricle or atrium [2]. Ischaemic mitral regurgitation (IMR) relates to the development of mitral regurgitation as a consequence of ischaemic cardiomyopathy, commonly following myocardial infarction (MI), with tethering of predominantly the posterior leaflet, and either Type IIIb Carpentier’s dysfunction with restricted leaflet motion in systole or, more rarely, due to Type I (isolated annular dilatation) or Type II (excessive leaflet motion) dysfunction. IMR may occur in either acute or chronic phases [1,3]. Using this definition allows for the distinction between IMR and chronic MR due to degenerative or rheumatic disease that might co-exist with concomitant coronary artery disease (CAD). IMR has been demonstrated to have poorer clinical outcomes, with the severity of MR having been reported to have direct implications on the development of heart failure and death [4].

In the primary percutaneous coronary intervention (PCI) era, the course of this disease has altered significantly, with earlier revascularisation being key in the prevention of permanent IMR, with improvements noted in long-term outcomes following acute MI [5]. Risk stratification is paramount following MI in order to identify those most at risk of developing chronic MR and heart failure. IMR may emerge at the early stage of MI, with the severity ranging from mild (frequently undiagnosed) to moderate and severe valve disease. Even transient ischaemia may lead to reversible IMR so long as the ischaemia is resolved. Fortunately, the most severe forms, such as papillary muscle (PM) rupture, are infrequent; however, mortality rates have been previously reported to be high [6]. These patients generally present cardiogenic shock and pulmonary oedema, and have a poor long-term prognosis even after surgical correction [7].

## 2. Epidemiology

IMR is prevalent during both the acute and chronic phases of MI, and is associated with a poor prognosis [5]. Fortunately, the incidence of mechanical complications following AMI have decreased over time. In older series, the incidence of mechanical complication post-AMI ranged from 1% to 5% [6,8,9,10,11,12,13,14,15]. However, in a large, randomized trial in which primary PCI was the reperfusion strategy, mechanical complications occurred in 0.91% of patients [8]. Therefore, in the PCI era, the prevalence of papillary muscle rupture after AMI is approximately 0.26% [8,9]. Despite this improvement, Bursi et al. [16] reported that up to 50% of patients with acute myocardial infarction (AMI) had evidence of MR upon echocardiography performed within 30 days after the ischemic event. In this study, almost 40% of patients had mild MR and 12% of patients in this study had either moderate or severe MR [16]. These authors also identified that patients with MR were older; had a greater number of comorbidities; and were more likely to be female, non-smokers, and have a higher Killip class (Killip classes III and IV) upon clinical presentation, in addition to having a lower EF [16]. In this study, no relationship was noted between the presence of IMR and the coronary artery territory involved [16]. Other studies have reported that IMR is more frequent in patients with anterolateral MI [17].

Papillary muscle rupture following AMI is exceedingly rare, with studies reporting an incidence of approximately 0.26% [8,9]. In other studies of patients requiring surgical intervention for acute papillary muscle rupture following acute MI, single vessel CAD was more frequent, with an incidence of between 23 and 44% [18,19].

## 3. Pathophysiology

The presence of acute MR primarily effects the pulmonary venous pressure and brings with it significant alterations in both the preload and afterload [20]. Pulmonary venous hypertension due to limitations in LV compliance results in a limited increase in end-diastolic volume [20]. This results in a preload dependant rise in LV stiffness, which also causes an increase in pulmonary venous pressures [20]. Subsequently, the total stroke volume increases due to LV preload reserves and the Frank starling mechanism [20]. In addition to this, the stroke volume is also effected by left atrial low pressure run off [20]. This initial increase in preload allows for a marginal degree of compensation; however, significant rises in both LV end diastolic and atrial pressures, and the resulting pulmonary oedema, are typical in this clinical scenario [20,21].

In a recent paper, Nishino et al. [22] described variations in mitral valve geometry in the setting of acute IMR in contrast with that of chronic IMR. The authors proposed that with acute IMR, in contrast with prior studies of chronic IMR, even a smaller degree of tenting in the setting of acute onset LV dysfunction could result in a loss of MV coaptation [22]. The subsequent reduction in leaflet area, in the absence of coaptation and in the context of larger hemodynamic burdens in the setting of acute MI, may be associated with clinically significant IMR [22]. Furthermore, the authors evaluated both the early systolic total leaflet area and the adaptation-related index, defined as the total leaflet area to annular area ratio. While the mitral annulus was remodelled in the setting of acute MI, there were no differences noted in the leaflet areas between the acute MR group versus both normal controls, and those with an acute MI without MR, with no differences observed in the extent of leaflet tenting [22]. It was also identified that pulmonary artery pressures were significantly higher in the acute IMR cohort, leading to the conclusion that the effects of elevated left atrial pressure on both annular dilatation and flattening contributed to poor leaflet coaptation [22]. Kim et al. proposed that leaflet tethering without the opportunity for compensatory increases in the mitral leaflet area resulted in acute IMR, implying an absence of valve adaptation in the acute phase [23].

Kimura et al. [24] also evaluated the physiopathology of MR in this subgroup of patients, using a 4D echocardiography dynamic model in a study conducted in Japan and the USA. Firstly, the authors reported that regardless of a comparable MI size, IMR was associated with a higher regurgitant volume (RVol), considerable changes in haemodynamic status, increased left ventricle (LV) volumes, and significantly reduced forward stroke volume, all of which are more pronounced in the presence of increasing IMR severity [24]. In addition, they noted in the acute setting of IMR that there was preservation of the annulus and functionality of the papillary muscles (PM) in contrast with loss of functionality in the chronic IMR setting [24]. Finally, and conceivably most importantly in this clinical setting, was the degree of angulation with its associated degree of tension on PM, with greater angulation resulting in disproportionate leaflet tension and regurgitation [24]. Although this is a dynamic state in acute IMR, it is present throughout the cardiac cycle [24].

The two processes that have been associated with the development of ischaemic MR following papillary dysfunction/rupture include (1) the disruption of the coronary blood flow and (2) changes in the left ventricular geometry [18,25,26]. The anterior papillary muscle (ALPM) is subtended by both the left anterior descending (LAD) and left circumflex artery (LCx), and is thus less vulnerable to ischaemia [25]. The posteromedial papillary muscle (PMPM) is supplied only by the posterior descending artery (PDA) with its long intramyocardial trajectory, and is thus more susceptible to ischaemia and rupture [25,27]. It has been reported that either complete or partial rupture of the posteromedial papillary muscle occurs between 6× and 12× more frequently because of its coronary blood supply [25,28,29]. The infarct size has no direct relation to the occurrence of total or partial rupture.

## 4. Clinical Presentation

The clinical presentation of patients with acute IMR can be broad, ranging from asymptomatic or milder symptoms, such as mild dyspnoea and coughing, to overt symptoms of congestive cardiac failure and haemodynamic compromise [25,30]. The acute development of secondary PM rupture due to acute MI is typically associated with acute congestive cardiac failure, in addition to significant haemodynamic compromise and poor clinical outcomes [25].

Upon physical examination, a pansystolic murmur in the mitral area radiating to the left axillary line is often present, albeit no direct relationship can be demonstrated between the degree of regurgitation and murmur intensity with cases of patients, with a poor cardiac output frequently having either a soft or absent murmur [25,31]. In addition to the typical murmur, classical signs of pulmonary oedema such as jugular venous distention and signs of respiratory distress might be evident [25,31,32].

## 5. Diagnosis

Echocardiography should be carried out immediately to confirm the diagnosis when there is a clinical suspicion of acute IMR. Beyond confirming the diagnosis, echocardiography also allows for a differential diagnosis of other mechanical complications, such as ventricular septal rupture (VSR). Transthoracic echocardiography (TTE) is well established as the initial diagnostic tool to identify papillary muscle rupture, with a diagnostic sensitivity of 65–85% [33]. As a result of the proximity of the ultrasound transducer to the mitral apparatus, transoesophageal echocardiography (TOE) is often used to better delineate the cause of MR and improves the diagnostic yield to between 95 and 100% (Figure 1). Additionally, with TTE, LV wall motion abnormalities, global left ventricular function, severity of leaflet prolapse, the presence of flail chords, and the integrity of papillary muscles can also be assessed. Colombo et al. [34] demonstrated that the sensitivity of both TTE and TOE for papillary muscle rupture improved significantly with the measurement of eccentric jets of MR using colour Doppler, by calculating the angle of the proximal MR jet and the plane of the mitral annulus. An angle of ≤47° on TEE and ≤45° on TTE established a sensitivity and specificity of 88% for flail mitral leaflets. The incremental improvement of echocardiogram technologies of 3D/dynamic 4D improved the diagnosis of acute IMR, thereby providing new insight regarding treatment options [24,35], as illustrated in Figure 2 and Figure 3. 

## 6. Treatment

### 6.1. Clinical Management: Pharmacotherapies and Mechanical Circulatory Support

Acute MR, with evidence of pulmonary oedema or cardiogenic shock, requires immediate action to improve haemodynamics and the emergent definitive intervention. Non-invasive positive pressure ventilation and high flow oxygen supplementation may be required in patients with pulmonary oedema, with endotracheal intubation and ventilation reserved for those with severe hypoxia or significant respiratory distress [36]. In patients with severe acute MR, pharmacological therapy primarily includes intravenous diuretics, vasodilators, and inotropes [36,37]. The primary goals of pharmacological therapy in this context are to reduce afterload and regurgitant fraction, and to improve LV contraction [37]. However, pharmacotherapy in these patients presents a significant challenge as alterations in preload or inotropy may in fact worsen MR or pulmonary congestion [38].

Vasodilators, including sodium nitroprusside, are most frequently used and are beneficial in reducing afterload, and aiding haemodynamic stabilization to facilitate definitive intervention [39,40]. Nitroprusside reduces both systemic vascular resistance and impedance of left ventricular ejection, thereby reducing MR [39,40]. Treatment with vasodilators may worsen hypotension, necessitating the use of vasopressors and or mechanical circulatory support (MCS). It was previously thought that in MR, vasopressors needed to be avoided due to the increase in afterload and its potential to increase regurgitant flow and pulmonary venous pressure [41,42]. However, vasopressors such as dopamine and dobutamine may lower blood pressure [41]. Secondly, the beta-1-adrenergic actions of both of these drugs favourably effect valvular competence by improving ventricular contractility and reducing systolic volumes. In addition to this, these drugs may decrease regurgitant volumes by curtailing ventricular systole [41]. Owing to its capacity to enhance cardiac contractility and reduce cardiac workload, levosimendan is an effective pharmacotherapeutic in this clinical context. In patients with reduced left ventricular ejection fraction, levosimendan has shown some benefit in the prevention of acute left ventricular deterioration both in patients undergoing surgical [43] and transcatheter mitral interventions [44].

In cases of acute cardiogenic shock, MCS has been described as a bridge to surgery. Intra-aortic balloon pump (IABP) is perhaps the most widely available device, and aids in reducing MR by reducing the afterload. It has shown promising results in small series and animal models; however, it does not offer significant improvements in cardiac output [38,45,46,47]. In addition, no significant differences in terms of operative mortality were observed in a recent meta-analysis between patients with or without pre-/peri-operative IABP [48]. Therefore, in patients requiring short-term MCS for stabilisation perioperatively, other devices can be considered. The Impella^®^ (Abiomed, Danvers, MA, USA) device directly unloads the ventricle and increases cardiac output by reducing retrograde flow across the valve and may allow for sufficient improvements in haemodynamics prior to surgery [38,49]. A TandemHeart Extra Corporeal Membrane Oxygenation (ECMO) (Cardiac Assist Inc., Pittsburgh, PA, USA) approach has also been described with some success [50]. These devices may aid in the stabilization of critically ill patients; however, prompt surgical intervention remains the gold standard of care.

### 6.2. Treatment: Surgical Management

Acute IMR is poorly tolerated, predominantly as a result of acute pulmonary oedema and haemodynamic compromise. A surgical approach is necessary to prevent biventricular failure leading to multiorgan dysfunction. In a retrospective study analysing the postoperative outcomes of emergent surgical intervention for acute MR by Lorusso et al. [51], the overall 30-day mortality was 22.5% [51]. A subgroup analysis determined that patients presenting with MI and acute endocarditis experienced significantly higher mortality rates, of 26.9% (*p* < 0.001) and 22.7% (*p* = 0.005), respectively [51]. With a multivariable analysis, independent predictors of higher mortality rates included ACS at presentation and concomitant CAD [51]. Prior studies have reported poorer survival rates in patients undergoing surgical intervention in patients with IMR in comparison with those with non-ischaemic MR [52,53]. However, recent studies have demonstrated that long-term patient survival is largely determined by baseline patient characteristics and comorbidities rather than an ischaemic etiology [53,54]. With regards to surgical techniques, the type of mitral valve surgery (repair versus replacement) did not alter the outcomes, even in the context of concomitant surgical coronary revascularization [51,55].

The timing of surgery is typically determined by the haemodynamic condition of the patient, with a reported median time to intervention (combined valvular and coronary artery bypass grafting) of 7 days [55,56,57,58,59]. However, in a series by Schroeter et al., more than half of the patients were operated on in the first 48 h after MI [59]. In a study by Kettner et al. [60], an early intervention strategy was recommended to improve 30-day mortality rates, which was largely driven by improvements in perioperative mortality [60]. Upon multivariate analysis, early progression to cardiogenic shock and unperformed surgery were the only independent predictors of 30-day mortality [60].

Patients with severe acute IMR due to PM rupture require rapid mitral valve surgery. While mortality rates in patients undergoing surgical intervention for acute IMR remain high [61,62], there have been reports of mortality in excess of 80% in the absence of intervention [63]. This cohort typically are in extremis and undergo surgical intervention in a critical state [7]. Despite being an uncommon pathology, there have been patient series published, with the majority undergoing MV replacement [51,57,59,61].

The reported mortality rates of these patients is considerably high, ranging from 12.5% to 39% [7,51,57,59]. Russo et al. reported a notable change in mortality with operative mortality decreasing from 28% prior to 1990, to 10% after 1990 (*p* = 0.09) [7]. Bouma et al. published a series of nine patients with no in-hospital mortality, all with partial PM rupture and relatively low pre-operative risk scores (only 1 patient with IABP), and reported a 90% MV repair rate with a 5-year survival of 83% [56].

#### 6.2.1. Mitral Valve Replacement

While the surgical techniques used in chronic IMR are widely variable [64,65], there is more of a consensus for the treatment of acute severe IMR, with valve replacement being the most common option in approximately 90% of cases [51,54]. The choice of the protheses is beyond the scope of this review; however, all aspects of this decision, such a as age, comorbidities, anticoagulation contraindications, lifestyle, or particular employment consideration, should be evaluated on an individual basis. It should be remembered that in certain circumstances, a direct discussion with the patient may not be feasible and in these circumstances, the patient’s next of kin should be part of the discussion.

Prior to the 1990s, sub valvular apparatus sparing MVR surgeries were favoured over the standard MVR technique with complete resection [66,67,68]. Following this, Yun et al. [69] published a small randomized trial comparing the total chordal structure preservation and preservation of only the posterior apparatus. The authors suggested that complete preservation was superior with regards to the ventricular remodelling and ejection fraction [69]. A further meta-analysis of bi-leaflet preservation included studies involving various preservation techniques, and it determined no significant differences between bi-leaflet preservation versus a posterior leaflet-only preservation approach, with both techniques resulting in a significant reduction in LV dimensions postoperatively [68,70]. As such, consideration should be given to bi-leaflet preservation so as to avoid a further reduction in LVEF in patients who exhibit extensive LV impairment [68].

An interesting concept was described by Gomes et al. [71], namely the Crossed Papillopexy. In this technique, the anterior leaflet undergoes central division with each half and its corresponding chordae tendineae complex is attached medially to the opposite commissure followed by the implantation of a prosthesis [71]. They advocated that this could improve LV remodelling and published promising early results.

However, some considerations should be taken into account. By reducing LV size, as in the case of bi-leaflet preservation, this may introduce the possibility of left ventricle outflow tract (LVOT) obstruction. Awareness of this possibility is needed to minimise this risk [68]. De Cannière et al. published a case report of a patient with “sequential” papillary muscle rupture in the acute phase following MVR, with the reported cause being excessive traction on the remaining papillary muscle heads after the application of the mitral leaflets [58]. With prior studies demonstrating similar results for total chordal preservation versus posterior leaflet preservation-only interventions [68], consideration should be given to the latter in order to mitigate against this traction [58].

#### 6.2.2. Mitral Valve Repair

Mitral valve repair, principally based on Carpentier’s reconstructive techniques [72], is the preferred strategy to treat chronic degenerative mitral valve regurgitation compared with mitral valve replacement, especially in the context of concomitant CAD, where improved perioperative mortality has been observed [73]. In this regard, in a recent meta-analysis by Massimi et al. considering 1851 patients evaluating early outcomes following surgical intervention for post AMI PMR, 18% of patients underwent repair. This study observed reduced operative risk in patients undergoing mitral valve repair versus replacement) (RR, 0.33; 95% CI: 0.14–0.79; *p* = 0.01) with increased risk in those with a complete PMR. The overall mortality rate was 21%, with no significant differences in operative mortality observed between patients with or without IABP or those requiring concomitant surgical revascularisation [48]. There are specific anatomical scenarios that infer a higher likelihood of success with regard to the durability of mitral valve repair, which include isolated posterior leaflet prolapse, rupture of a chordae adjacent/adherent to the posterior leaflet, and commissural prolapse [74]. Importantly, these anatomic variances are more likely to be seen in patients with chronic degenerative MR as opposed to acute IMR. In the setting of acute IMR, these analyses have not been performed in a sufficiently large prospective trial.

As discussed earlier, this is not the preferred treatment strategy in these patients. However, there are some reports that valve preservation can be achieved, even in patients with PM rupture. Sultan et al. [57], in a small series of 24 patients, described successful mitral valve repair in 30% of patients and this was primarily achieved in cases of incomplete PM rupture. Similar findings were identified in a small study of nine patients, all with partial PM rupture, all of whom underwent surgical valve repair [56]. In this study, patients were more haemodynamically stable, with only one patient requiring an IAPB, and in the majority of cases, the surgery was not emergent or salvage, with only two cases in the first 7 days after MI. Another notable finding of this series was that P2 prolapse was primarily treated with a classical quadrangular resection and annuloplasty ring [56].

Mitral valve repair remains a challenging valve surgery and, unsurprisingly, individual surgeon volume and high volume centres have a direct impact on improved outcomes [75]. Various surgical techniques have been described; however, head-to-head comparisons are lacking. Despite this, ring annuloplasty is the most widely accepted technique for MV repair. Following mitral valve repair, concerns have been raised regarding the development of systolic anterior motion (SAM) in the immediate post operative period. In a large study by Ashikhmina et al., post operative SAM was observed in 13% of cases [76]. Anatomic factors associated with SAM include a height discrepancy between the posterior and anterior mitral leaflet, small left ventricular end-systolic volume, and bi-leaflet prolapse. It was noted, however, that in the majority of cases, conservative management was possible [76].

Emerging techniques, such as NeoChord echo-guided transapical repair, have been used in chronic degenerative MR with good outcomes in medium- and long-term follow up [77]. The translation of these repair techniques in the context of acute IMR are anticipated. Other novel techniques, such as mitral valve translocation surgery in IMR, have been developed in an attempt to overcome the long-term durability concerns of traditional mitral valve repair techniques [78]. The early experience with this method is limited to chronic IMR; however, further studies are warranted before conclusions relating to its efficacy in the setting of acute IMR can be made. Similarly, minimally invasive surgical techniques, such as the HARPOON™ beating heart mitral valve repair system (Edwards Lifesciences, Irvine, CA, USA), have been studied in primary MR; however, to date, they have not been applied to the acute IMR setting [79].

### 6.3. Treatment: Percutaneous Management

While prompt surgical intervention is recommended for acute symptomatic severe mitral regurgitation, it is frequently denied as a result of prohibitive surgical risk [31,80]. There are emerging data to support the use of Transcatheter MitraClip^®^ device (Abbott Vascular, Santa Clara, CA, USA) implantation in the setting of significant symptomatic MR, and in patients in whom surgical intervention has been declined due to excessive risk [80]. The first two cases were described in 2014 [81,82], and following this, other case reports have reported procedural success in this clinical scenario [83]. Overall, there is a paucity of data regarding the use of the MitraClip device in this rare clinical setting, and there is a notable absence of randomized clinical data. Several registries have been recently published; however, these reviewed experience with acute IMR and treatment with MitraClip. A small Spanish registry reported their experience with MitraClip in the setting of AMI [83]. The patients had a median EuroScore of 29.1% and acute procedural success was observed in all cases with no major complications [83]. One patient died from multiorgan failure during the admission, with the remaining patients followed up for a median of 317 days [83]. At follow up, all patients had MR ≤ 2+ with NYHA functional class ≤ II [83]. The national Israeli MitraClip Registry (IMCR) [84] was a multicentre retrospective registry involving nine Israeli centres, with additional data from centres in Zurich, Toronto, and Athens. This was a study of 20 patients (8 IMCR and 12 others) with severe IMR within 90 days following MI without damage or rupture of the PM or chordae [84]. In all but one, the MitraClip was successfully implanted, and the early mortality was found to be 5% [84]. One patient suffered sudden death 3 weeks after discharge, and no late deaths between 3 and 88 months were observed [84].

Estévez-Loureiro et al. published the first prospective, multicenter study, from the European Registry of MitraClip in Acute Mitral Regurgitation following acute myocardial infarction (EREMMI) [85], where 44 patients with acute IMR after a median of 18 days after MI and a high median EuroSCORE II of 15.1 were treated with the MitraClip [85]. In this study, 86.6% of patients had the MitraClip successfully implanted, with good haemodynamic improvement [85]. The 30-day survival was 90.9% and 81.8% at 6 months [85]. Finally, Haberman et al. [86] published a retrospective international registry data study that included 471 patients, of which 266 were treated conservatively, 106 were treated with surgery, and 99 patients had percutaneous mitral valve edge to edge repair [86]. Those patients who underwent mitral valve interventions were in poorer clinical states, yet their in-hospital and 1-year mortality rates were lower [86]. There were also differences noted between the surgical and percutaneous groups, with the surgical group experiencing significantly higher in-hospital and 1-year mortality [86]. The STS-ACC TVT Registry [87] reported data for patients undergoing TMVR and TEER for various indications, including those directly approved by the FDA and also secondary MR in patients with prohibitive surgical risk treated with TEER. This registry reported in-hospital, 30-day, and 1-year mortality rates for the two interventions (mortality at 1 year for TEER 16.4% versus TMVR 16.3%). However, there has been no head-to-head comparison of these two percutaneous techniques in the setting of acute IMR.

While these data suggest that it is safe and feasible to treat IMR with MitraClip in these patients, further studies are warranted before we can make definitive conclusions regarding its use in this high-risk clinical context. It should be remembered that surgical intervention should remain the gold standard of care in acute IMR, with the consideration for MitraClip reserved for patients with prohibitive surgical risk, and only after careful consideration and discussion among the heart team.

## 7. Conclusions

Acute IMR is a serious complication following acute MI. The occurrence of mechanical complications, such as papillary muscle rupture, are increasingly rare. The surgical success rates depend on the surgical team’s experience and expertise, the timing of the procedure, and the clinical state of the patient. Generally, the acute course of IMR is associated with significant haemodynamic compromise, with a greater number of patients requiring MCS and ventilatory support, and urgent or salvage intervention. Surgical intervention remains the gold standard intervention in this context.

The development of percutaneous edge-to-edge techniques might offer an alternative in those patients in whom surgical risk is considered prohibitive. MitraClip is now widely accepted in suitable candidates as an alternative to mitral valve repair for high-risk patients with chronic MR, and while emerging registry data appear to support its use in the context of acute IMR in patients with excessive surgical risk, further randomised data are urgently needed. In addition to this, it is clear that careful assessment by the Heart Team with tailored peri-procedural planning in these challenging cases is warranted prior to embarking on an intervention.

## Figures and Tables

**Figure 1 jcm-11-05526-f001:**
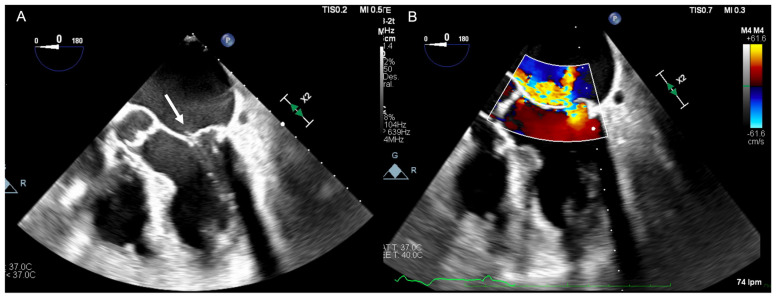
(**A**) Transoesophageal echocardiographic four chamber mid oesophageal view illustrating a flail posterior mitral valve leaflet with ruptured chordae (white arrow). (**B**) Colour Doppler demonstrating an eccentric jet of severe mitral regurgitation.

**Figure 2 jcm-11-05526-f002:**
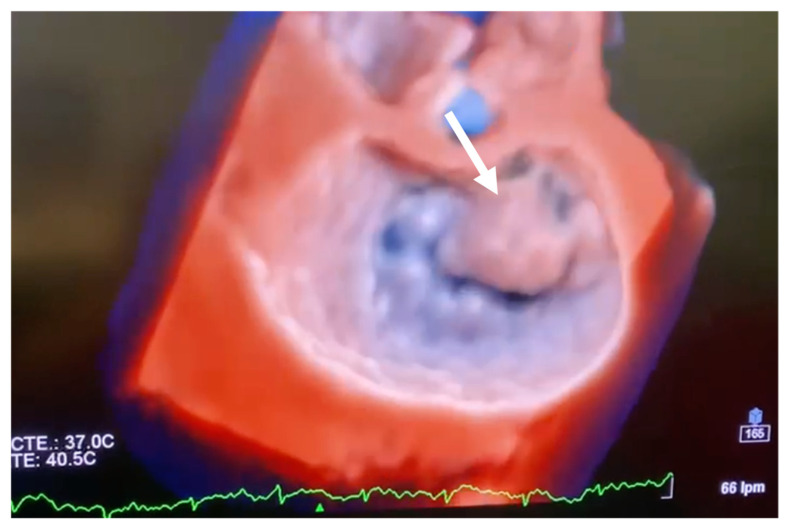
Transoesophageal echocardiographic short axis 3D view (surgical view) illustrating anterior leaflet prolapse (white arrow).

**Figure 3 jcm-11-05526-f003:**
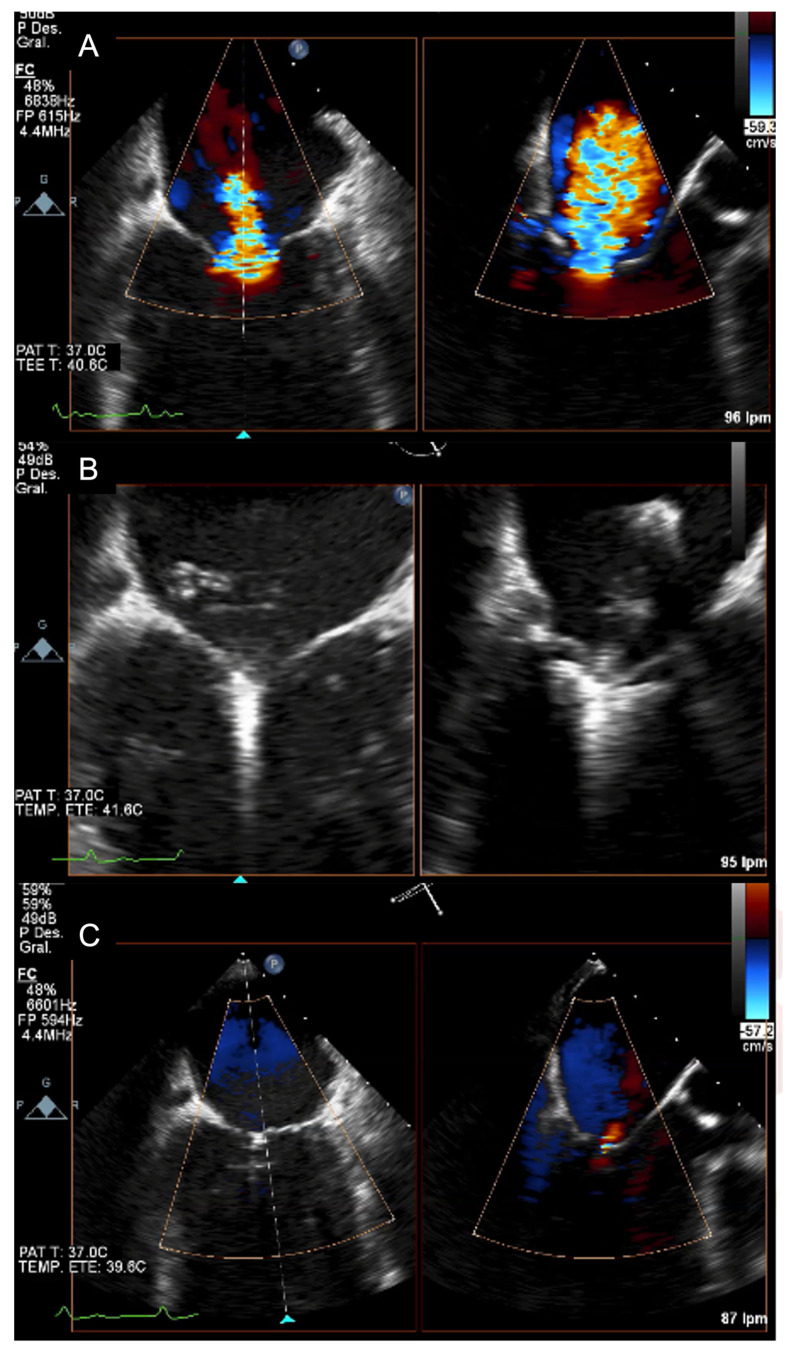
(**A**) Transoesophageal echocardiographic: colour Doppler demonstrating severe mitral regurgitation in a patient with acute MR. (**B**) Transoesophageal echocardiographic: grasping with a MitraClip XT device. (**C**) Transoesophageal echocardiographic: final result: trace MR.

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
