# Peer review of "Acute Ischaemic Mitral Valve Regurgitation"

_jcm, 2022, doi:10.3390/jcm11195526_

Round 1

Reviewer 1 Report

The manuscript discusses Ischemic mitral insufficiency, its pathophysiology and its treatment. It describes how anatomic and functional features related to myocardial ischemia/infarction can result in mitral insufficiency. It briefly discussed the age, sex, and smoking characteristics of those likely to develop ischemic mitral insufficiency.  It also discusses treatment options.

Overall, the manuscript has a good general approach to understanding ischemic mitral insufficiency. Unfortunately, it seems a bit superficial in most aspects. While it covers basic components as to the etiology of MR and the impact of coronary anatomy on ischemic MR, there is scant discussion of important issues such as:

1) factors favoring valve replacement rather than repair; 

2) neocord/ring surgery and the predictors of success are scant. 

The information on newer treatment strategies seems lacking. There is no discussion on:

1) Transcatheter mitral valve replacement, its efficacy, and its indications with compared with transcatheter edge-to-edge mitral valve repair

2) Harpoon beating heart apical mitral valve repair

3) mitral valve translocation surgery

4) factors predisposing to post-treatment systolic anterior motion of the mitral valve. 

Additionally, there are many run-on sentences and inappropriate words used. As an example, Levosimendan is an Effective (not Affective) agent

Reviewer 2 Report

Dear Authors,

the paper is interesting,  i think can be usefull to review the therapeutical approach also with indication to IABP ( intra aortic balloon pump) use and  a better definition  of the pharmacological treatment .

I can send some scientific article about IABP use in acute mitral valve regurgitation

1. shock 003 Apr;19(4):334-8. doi: 10.1097/00024382-200304000-00007.

2. Int J Surg Case Rep . 2013;4(1):5-6. doi: 10.1016/j.ijscr.2012.09.004. Epub 2012 Sep 26.

Reviewer 3 Report

This article focuses on acute ischemic MR with onset immediately following a myocardial infarction.  While the article briefly mentions chronic IMR due to long-term coronary artery disease, this is not the focus of the review.  I believe the authors could do a better job in the introduction defining the broad concept of IMR, as well as devote a small paragraph to chronic IMR that occurs without acute MI but in the presence of chronic coronary artery disease, as that is the more common type of IMR.  This would reduce confusion later in the manuscript where there are references to chronic IMR, where it is not clear if it refers to post-MI chronic IMR or traditional IMR occurring in the setting of coronary artery disease.

The discussions on papillary muscle rupture seem forcibly interspersed within the discussion and disrupt the flow of the manuscript.  Since PM rupture is a rare occurrence, I would recommend that the paragraphs be placed at the same location within the section e.g. at the end of the section, befitting a less-common event.

Much is made of the ‘PCI era’ literature, but this is not done with a comparison to any time prior to the widespread acceptance of PCI.  If the goal is to indicate that due to PCI, the outcomes following MI with acute MR are better, then this should be more clearly stated in the overall tone of the manuscript.

The use of MitraClip in patients with papillary muscle rupture seems like a poor choice.  I can understand its use in patients with acute or chronic IMR who have prohibitive surgical risk, but its recommendation for use in patients with PM rupture should be better explained/justified

Round 2

Reviewer 2 Report

I believe that you have improved the quality of your article

Reviewer 3 Report

Appropriate for acceptance as revised.